# Dietary Intake by Food Source and Eating Location in Low- and Middle-Income Chilean Preschool Children and Adolescents from Southeast Santiago

**DOI:** 10.3390/nu11071695

**Published:** 2019-07-23

**Authors:** Natalia Rebolledo, Marcela Reyes, Camila Corvalán, Barry M. Popkin, Lindsey Smith Taillie

**Affiliations:** 1Department of Nutrition, Gillings School of Global Public Health, University of North Carolina at Chapel Hill, Chapel Hill, NC 27599-7400, USA; 2Institute of Nutrition and Food Technology (INTA), University of Chile, Santiago 7830490, Chile; 3Carolina Population Center, University of North Carolina at Chapel Hill, Chapel Hill, NC 27516-2524, USA

**Keywords:** child diet, adolescent diet, food source, eating location, Latin America, away-from-home food, school food, fast food, sugar-sweetened beverage

## Abstract

Background: Food source and eating location are important factors associated with the quality of dietary intake. In Chile the main food sources and eating locations of preschool children and adolescents and how these relate to dietary quality are unknown. Methods: We analyzed 24 h dietary recalls collected in 2016 from low- and middle-income Chilean preschool children (3–6 years, n = 839) and adolescents (12–14 years, n = 643) from southeastern Santiago. Surveys collected the food source and eating location for each food reported during the recall. We estimated the mean intake of calories and key nutrients of concern, such as saturated fats, total sugars, and sodium, by food source and eating location. Results: Foods obtained and eaten at home contributed the greatest proportion of total calories and the key nutrients of concern. Foods obtained at home tended to have lower caloric densities but higher sugar and sodium densities than foods obtained away from home in both age groups. With regard to location, for preschool children foods consumed at home had lower caloric and sugar densities than foods eaten at school, while for adolescents foods consumed at home had lower caloric, saturated fat, and sugar densities than foods eaten at school. For both children and adolescents, home was the primary source of sugar-sweetened beverages (SSBs) calories. SSBs were important calorie contributors among foods across all settings, but the highest absolute amount of calories from these beverages was consumed at home. Conclusions: While most of Chilean youths’ calories and key nutrients of concern are obtained and consumed at home, these foods tended to have lower caloric densities than foods obtained and consumed away from home. Home was the main food source for SSBs, but the relative consumption of these beverages was high in all eating locations. More research will be needed to inform and evaluate policies and interventions to improve children’s dietary quality across settings.

## 1. Introduction

As in other Latin American countries [1,2,3], overweight and obesity have increased rapidly in Chile over the last 30 years [4,5]. According to the 2016–2017 Chilean National Health Survey, the combined prevalence of overweight and obesity was 74.2% among individuals 15 years and older [6], and more than 50.0% of preschool children were overweight or obese [7]. In the last 20 years the Chilean diet has changed, with a decrease in consumption of legumes, fruits, and vegetables [8] and an increase in consumption of sugar-sweetened beverages (SSBs), ready-to-eat foods, and processed and ultra-processed foods [8,9]. According to the Chilean National Dietary Survey of 2010, the dietary share of ultra-processed foods reached almost 30.0% of total daily calories [10]. Although little research has investigated the determinants of obesity specific to Chile, the increase in obesity worldwide has coincided with changes in dietary behaviors, such as increased snacking, increased away-from-home food intake, larger portion sizes [11], and decreased time spent in food preparation [12]. A growing body of evidence has found that food source (i.e., where food is obtained) and eating location (i.e., where food is consumed) are important factors associated with the quality of dietary intake [13,14,15,16,17,18,19,20], a significant contributor to excess weight.

Studies in the United States have shown that while foods obtained at home contribute the largest proportion of children’s daily calories, foods eaten at home are not necessarily of better nutritional quality than foods eaten at other locations [15]. Furthermore, foods children and adolescents obtain from stores and schools are not lower in solid fats and added sugars (empty calories) than foods obtained from fast food restaurants [17]. Similarly, among Mexican children foods eaten at home contribute the most total calories and calories from solid fats and added sugars [18]. In contrast, among Brazilian children, lunches consumed at home are of better nutritional quality (according to the amounts of total fats and saturated fats) than lunches consumed away from home [21]. In Chile children’s and adolescents’ main food sources and eating locations and how these relate to dietary quality are unknown. In addition, it is important to characterize how the consumption of calories and other key nutrients of concern, food groups, and nutrient densities varies by food source and eating location. Identifying which food sources and eating locations are associated with less healthful food intake will help policy makers appropriately target the dimensions of the food environment for policies and interventions to improve children’s and adolescents’ diets.

Using dietary data from the 2016 waves of two Chilean cohorts from Southeast Santiago, this study aimed to examine preschool children’s and adolescents’ daily intakes of key nutrients of concern (calories, sugars, saturated fats, and sodium) by food source and eating location, to examine preschool children’s and adolescents’ daily calories from food and beverage groups by source and eating location, and to determine the nutrient densities of foods by source and eating location.

## 2. Materials and Methods

### 2.1. Participants

We used data from the 2016 waves of two longitudinal cohort studies: the Growth and Obesity Cohort Study (GOCS), which consists of adolescents 12–14 years old, and the Food Environment Chilean Cohort (FECHIC), which consists of preschool children that are 3–6 years old. Both cohorts are composed of people of low- and middle- income based on the income levels of their neighborhoods of residence. The GOCS began in 2006 and included preschool children aged 2.6 to 4.0 years old who were attending any of the 54 preschools belonging to the National Association of Day Care Centers in southeastern Santiago, Chile. These children were 12–14 years old during the 2016 study wave. The inclusion criteria for the GOCS were children who were singleton term births, had birth weights between 2500 and 4500 g, and were free from conditions that affect growth, such as genetic and metabolic diseases. Further details on recruitment procedures have been previously described [22]. The FECHIC was initiated in 2016, and the recruitment methods were similar to those of the GOCS. The FECHIC included preschool children from 50 of the 55 public schools that included preschool (i.e., school programs for children 3–4 years old) in southeastern Santiago. (Note, the remaining 5 public schools chose not to participate in this study.) The FECHIC inclusion criteria were children who were singleton births who were free from gastrointestinal diseases that affect food consumption. Additionally, the mother had to be responsible for household food purchases, the child’s primary caretaker, and free of any mental disability. For both cohorts we obtained informed consent from the parents or guardians of the participants. We also obtained informed assent from the adolescents. We conducted the study in accordance with the Declaration of Helsinki, and the protocol was approved by the Ethics Committee of the Institute of Nutrition and Food Technology (INTA) of the University of Chile and the Institutional Review Board of the University of North Carolina at Chapel Hill (UNC-CH).

### 2.2. Dietary Data

For this analysis we used dietary data collected in the first semester of 2016 for consistency between the cohorts. Trained dietitians collected 24 h dietary recalls using SER-24 software developed at the INTA following the US Department of Agriculture (USDA) multiple-pass method [23]. Foods and beverages were linked to the most similar food or beverage in the USDA Food Composition Databases [24]. Additionally, we used the Photographic Atlas of Chilean Foods and Typical Preparations, which was validated in the Chilean National Dietary Survey [25], to identify the serving sizes of common Chilean foods and beverages according to glasses, mugs, or plates to assist participants in estimating the portions they consumed. In the cases of sugars, oils, and other condiments, we used cups or spoons to measure portions. During the in-person interview, the child or adolescent was accompanied by one caretaker (a parent or a guardian) who was aware of that child or adolescent’s food intake during the previous day. The adult was the primary respondent to the 24 h dietary recall for a preschool child, who complemented the information about foods consumed when the primary caretaker was not present. Adolescents were the primary respondents to their 24 h dietary recalls, and adults helped them remember which foods and beverages they had consumed in the previous 24 h.

The surveys collected dietary data for 962 FECHIC participants and 770 GOCS participants. To capture the intake on school days, we excluded dietary recalls performed during weekends or holidays. A second 24 h dietary recall was collected for 20.1% of the FECHIC participants and 13.5% of the GOCS participants. For these participants, we included the first recall collected on a normal weekday (Monday through Friday). Additionally, we excluded participants who were sick and who reported their recalls as a special occasion (e.g., a birthday). Our total analytic sample was 643 participants (82.3%) from the GOCS and 839 participants (87.2%) from the FECHIC.

### 2.3. Food Source and Eating Location

For each food or beverage consumed we asked the participant or the caretaker for the food source (i.e., where the food was acquired) and the eating location (i.e., where the food was consumed). We classified the food sources as (1) home, including products made or present at home or purchased from grocery stores, convenience stores, or supermarkets, or (2) away from home, including products obtained or purchased at school (i.e., food provided by the school meal program or purchased on school property), restaurants, fast food establishments, or other sources. For the source variable, we combined school with other away-from-home sources, because we were unable to differentiate foods obtained through the school meal program from those obtained from outside vendors (e.g., kiosks) at school. We classified eating location as (1) home if the foods or beverages were consumed at home; (2) school if the foods or beverages were consumed at school; and (3) other away from home if the foods or beverages were consumed at another person’s home (i.e., friend’s, grandparent’s), at a food court, at a cinema, at a restaurant, in the street, on transportation, or at another location.

### 2.4. Food and Beverage Groups

Our team grouped foods and beverages based on their nutritional compositions, their typical eating patterns in Chile, and the reported methods of preparation following a previously developed food grouping system [26]. We classified food and beverage groups as basic if they contribute to the intake of essential nutrients (e.g., cereal-based foods, eggs, legumes, fruits, and vegetables) or nonbasic if they provide high caloric and low nutrient densities (e.g., grain-based desserts, salty snacks, sugary beverages) [27]. The food and beverage groups we used are described in Appendix A. We classified mixed dishes with a standardized cooking recipe in SER-24 as a whole dish (e.g., we classified *Arroz graneado*, a rice-based dish with vegetables, as a mixed dish, grain based). We classified mixed dishes without a standardized cooking recipe in SER-24 by disaggregating their components and classifying them accordingly (e.g., we classified *Arroz perla*, white rice, as a cereal-based food; oil as oils and fats; and salt as salt and seasonings).

### 2.5. Nutrient Density by Food Source and Eating Location

We calculated nutrient density per 100 grams [g] and per 100 calories for each food source and eating location. For each 100 g of a specific food, we calculated the caloric (kilocalories [kcal] per 100 g), saturated fat (g per 100 g), total sugar (g per 100 g), and sodium (milligrams [mg] per 100 g) densities. For each 100 calories of a particular food group, we calculated the percentage of calories from saturated fats (percentage of total calories from each source or location), the percentage of calories from total sugars (percentage of total calories from each source or location), and the sodium density (mg per 100 kcal). We focused on key nutrients of concern such as calories, saturated fats, total sugars, and sodium because recent Chilean health and nutrition policies were designed to reduce intake of these nutrients as a strategy for obesity prevention [28,29] and because health and nutrition organizations have recommended limiting their consumption [30,31].

### 2.6. Statistical Analyses

We used STATA (version 14.2, 2016, StataCorp, College Station, TX, USA) to perform our analyses. We used a single 24 h dietary recall to compute the proportions of the preschool children and adolescents obtaining food from different sources and consuming food at different locations and the mean intake of calories, saturated fats, total sugars, and sodium by food source and eating location. We present the results per capita (i.e., the mean for the entire study population) and per consumers (i.e., the mean for the subsample consuming at least one food or beverage item from a given food source or eating location). For food group analyses we identified which food groups were the main contributors of daily calories, and we calculated the percentage of calories they contributed using the total calories consumed at each food source or eating location as the denominator. We assessed the normality of the distribution of the nutrient density variables by food source and eating location with the Shapiro-Wilk test. Since the distribution of the nutrient density variables differed from a normal distribution, we assessed differences in nutrient densities by food sources and eating locations for each child with the Wilcoxon signed rank test. Home was both the referent food source and the referent eating location. We present the median and the interquartile range (IQR) for the nutrient density variables. We defined statistical significance at *p* < 0.05.

### 2.7. Sensitivity Analyses

Because of the right skew of the distributions for saturated fats and sodium, we examined the mean intake of these nutrients with and without outliers (>99.5th percentile). In an additional sensitivity analysis, we included only 24 h dietary recalls collected on weekends and holidays to determine if there were differences in the proportions of calories and key nutrients of concern obtained from each food source and consumed at each eating location.

## 3. Results

### 3.1. Characteristics of Participants

The sociodemographic characteristics, mean intake of key nutrients of concern, and percentage of consumers by source and location are in Table 1. The mean age was 4.3 (± 0.5) years for preschool children and 13.2 (± 0.5) years for adolescents. The mean per capita daily calorie consumption was 1257 (± 13.6) kcal/day for preschool children and 1826 (± 24.9) kcal/day for adolescents. During the recall day all of the preschool children and almost all of the adolescents obtained food from home at least once, while 59.2% and 72.3%, respectively, obtained at least one food item from an away-from-home source. Similarly, almost all of the preschool children and adolescents consumed food at home, and three-quarters of them consumed food at school. Less than half of the study sample consumed food at other locations.

### 3.2. Food Sources

In both preschool children and adolescents the per capita analysis showed that the greatest source of calories was home, providing 82.8% and 77.6%, respectively, of total daily calories (Figure 1). The other analyzed nutrients followed a similar pattern, with the highest contributions coming from at-home sources. Per consumer results are in Appendix A.

### 3.3. Eating Locations

The per capita analysis showed that home was where preschool children and adolescents ate most of their total daily calories (66.3% and 62.8%, respectively), followed by school (22.3% and 26.6%, respectively) (Figure 2). This pattern was also consistent for intake of key nutrients of concern. Per consumer results are in Appendix A.

### 3.4. Nutrient Density by Food Source

Table 2 shows the calories and nutrient profiles by food source. The preschool children results per 100 g showed that foods obtained at home were less caloric dense, contained similar amounts of saturated fats, and contained higher amounts of total sugars and sodium compared with foods obtained away from home. Results per 100 calories followed the same pattern.

The adolescents results per 100 g showed that foods obtained at home were less caloric dense, contained lower amounts of saturated fats, and contained higher amounts of total sugars compared with foods obtained away from home. When analyzed per 100 calories, foods obtained at home had higher sodium densities than away-from-home foods, but we found no statistically significant differences between food sources in the percentages of calories from saturated fats and total sugars.

### 3.5. Nutrient Density by Eating Location

Table 3 shows the calories and nutrient profiles by eating location. Among preschool children results per 100 g showed that foods eaten at home were less caloric dense, contained lower amounts of total sugars, and contained higher amounts of saturated fats and sodium than foods eaten at school. Results per 100 calories followed the same pattern. Foods eaten at home had similar caloric densities and higher amounts of saturated fats and sodium per 100 g compared to foods eaten at other away-from-home locations. Results per 100 calories showed that foods eaten at home had lower percentages of calories from total sugars and higher amounts of sodium compared to foods eaten at other away-from-home locations.

Among adolescents, foods eaten at home were less caloric dense, contained lower amounts of saturated fats and total sugars, and contained higher amounts of sodium per 100 g compared to foods eaten at school. Foods eaten at home were less caloric dense, contained lower amounts of total sugars, and contained higher amounts of sodium per 100 g compared with foods eaten at other away-from-home locations. Results per 100 calories followed the same pattern in both school and other away-from-home locations.

### 3.6. Food Groups Obtained from Each Food Source

For preschool children, the top calorie contributors obtained at home were SSBs, other food groups, cereal-based foods, mixed dishes, and grain-based desserts (Table 4). In contrast, the top calorie contributors obtained away from home were mixed dishes, empanadas and sandwiches, unsweetened dairy drinks and substitutes, grain-based desserts, and other food groups.

For adolescents the top calorie contributors obtained at home were cereal-based foods, grain-based desserts, mixed dishes, SSBs, and other food groups. The top calorie contributors obtained away from home were mixed dishes, empanadas and sandwiches, grain-based desserts, SSBs, and cereal-based foods.

### 3.7. Food Groups Consumed at Each Eating Location

For preschool children the top calorie contributors consumed at home were other food groups, mixed dishes, cereal-based foods, and SSBs. The top calorie contributors consumed at school and other locations were mixed dishes, SSBs, grain-based desserts, and other food groups. Dairy products and dairy substitutes and unsweetened dairy drinks and substitutes were also top contributors at school, while at other locations cereal-based foods and sweets and non-grain-based desserts were also top contributors.

For adolescents the top calorie contributors consumed at home were cereal-based foods, mixed dishes, SSBs, and grain-based desserts. The top calorie contributors consumed at school and other locations were grain-based desserts, cereal-based foods, SSBs, and empanadas and sandwiches. Mixed dishes were also top contributors at school, while at other locations fast foods were also top contributors.

Both age groups consumed disproportionately greater amounts of cereal-based foods at home and disproportionately greater amounts of mixed dishes and grain-based desserts at school. At other locations, both groups consumed disproportionately greater amounts of grain-based desserts, cereal-based foods, sweets and non-grain-based desserts, and salty snacks (Table 5).

### 3.8. Sensitivity Analyses

The results remained consistent after we excluded the outliers for saturated fats and sodium at the 99.5th percentile (Appendix A). When we repeated the analyses on 24 h dietary recalls collected during weekends, home remained the main food source and eating location for calories in both age groups (Appendix A). In preschool children and adolescents, the per capita analysis showed that 86.3% and 84.3%, respectively, of calories were obtained at home (Appendix A). Similarly, home was where preschool children and adolescents ate most of their total daily calories (72.8% and 71.2%, respectively), followed by other locations (26.6% and 28.2%, respectively) (Appendix A). Results for saturated fats, total sugars, and sodium followed the same patterns.

## 4. Discussion

To our knowledge this is the first study that reports how the intake of calories, key nutrients of concern, and food and beverage groups varies by food source and by eating location among preschool children and adolescents in southeastern Santiago, Chile. We found that they obtained and consumed most calories, saturated fats, total sugars, and sodium at home. Foods obtained at home were lower in caloric density but had higher total sugar and sodium densities when compared with away-from-home foods. However, foods eaten at home were lower in both caloric and total sugar densities, especially when compared with foods eaten at school. Foods consumed at school included disproportionate amounts of mixed dishes (provided by the school meal program) and nonbasic foods and beverages (i.e., foods and beverages typically high in discretionary or empty calories) [17]. Preschool children and adolescents obtained and consumed most of their SSB calories at home. However, this occurred because both age groups obtained and consumed most food at home. Relative to the total amount of calories consumed at home, school, and other away-from-home locations, SSB intake was high in all locations.

Our findings indicated that the overall amount of calories from away-from-home sources was low. Chilean preschool children and adolescents obtained only 17% and 22%, respectively, of their daily calories from away-from-home sources. This was lower than US children and adolescents, who obtained 24% and 35%, respectively, of their daily calories from away-from-home sources [15]. Chilean children and adolescents did not obtain or consume large amounts of calories at school, fast food establishments, restaurants, or other away-from-home locations during weekdays. Our sensitivity analyses indicated that consumption at other locations increased over the weekend but that home was still the main food source and eating location. The low consumption of food away from home suggested that the amount of calories obtained from fast food establishments and restaurants was low, indicating that the nutrition transition in Chile has not advanced with regard to where food is sourced [11]. However, this might change in the future, because purchases per capita in fast food outlets are increasing [32].

The foods and beverages obtained or consumed at home came from similar food groups for both preschool children and adolescents. The largest calorie contributors at home were basic foods, such as cereal-based foods and mixed dishes. However, at-home foods also included a high amount of calories from grain-based desserts and SSBs, consistent with previous literature showing that Chileans’ intake of SSBs is very high [33,34]. These patterns are also consistent with findings in the United States, where stores (which in this study were categorized as at-home sources) were the largest contributors of SSBs for children [35]. Interestingly, the dietary intake data used in this study were collected after a 2014 nationwide tax on SSBs increased the tax rate on high-sugar beverages from 13% to 18% but excluded sugar-sweetened milks. The current results suggest that even after the implementation of this tax SSB intake among preschool children and adolescents remained high. This is consistent with an evaluation of Chile’s SSB tax that found that SSB purchases declined by only 3% after the tax was implemented [36]. The fact that we found such high levels of SSB intake at home (which includes both homemade and store-bought beverages) suggests that additional policies and interventions in the home and store food environments are needed to reduce intake of these beverages.

Among foods consumed at other locations the largest contributors of calories for preschool children were grain-based desserts, SSBs, and sweets and non-grain-based desserts, compared to empanadas and sandwiches and fast foods for adolescents. These results are similar to studies of Mexican children [18] and Brazilian adults [37] that also found that snack foods and desserts made up the greatest proportion of calories from foods consumed away from home. Additionally, we found that foods consumed at other locations had higher percentages of calories from total sugars for preschool children and adolescents and were more caloric dense for adolescents. Even though we found that foods consumed at home contributed the majority of calories in Chile, away-from-home eating may increase in the future. Future research will be needed to understand the contribution of away-from-home eating to excess consumption of key nutrients of concern and nonbasic foods.

Our results suggested that foods and beverages eaten at school had the least healthy nutrient profile, which was likely due to the food groups relatively most consumed at school. In both age groups the highest calorie contributors consumed at school were mixed dishes and nonbasic foods, including grain-based desserts and SSBs. School was the second-top contributor of absolute SSB calories for both preschoolers and adolescents, and for preschoolers the relative contribution of SSB calories to total calories consumed at a particular location was highest at school. Moreover, foods consumed at school had higher caloric and total sugar densities than foods eaten at home in both age groups. One interesting finding was that mixed dishes contributed more than 20% of the calories consumed at school. A possible explanation is that low- and middle-income preschool children and adolescents receive meals from the school meal program. Unfortunately, the dietary intake survey did not differentiate between foods obtained through the school meal program, those obtained at school from outside vendors, and those brought from home. There may be nutritional differences between foods obtained from these sources. A previous study in Chilean children showed differences in the food groups children consumed depending on whether they brought their foods from home or bought them at school [38]. Children who had money to buy food at schools mostly bought sweet snacks, juice, ice cream, and salty snacks, while children who brought food from home mostly brought juice, fruit, and yogurt [38]. Future research is needed to disentangle the nutritional quality of foods from the school meal program, foods consumed at school but brought from home, and foods purchased from kiosks or other vendors at school. Given that food preferences and eating behaviors develop throughout childhood and adolescence [39,40], policies and interventions to modify the eating environments at schools can improve the overall nutritional quality of diets and encourage healthy food preferences.

Notably, the dietary data we used were collected prior to the implementation of Chile’s 2016 regulation that banned the sale and marketing of foods and beverages high in calories, sugars, sodium, and saturated fats at schools; required front-of-package warning labels; and placed marketing restrictions on these products [29]. Future research is needed to understand whether the sales ban improved the nutritional quality of foods obtained at schools and whether the regulations improved the nutritional quality of foods consumed across sources and locations.

This study had several limitations. First, because the frequency of dietary recalls was disproportionately greater on weekdays than on weekends, our main analysis focused on dietary intake during weekdays. While our sensitivity analysis found that weekend eating patterns were similar, future research is needed to understand the overall usual intake of nutrients and food groups across food sources and eating locations, including both weekdays and weekends. Second, we defined income level by residence area and not by family income. It would be useful to understand whether eating patterns by source and location differ by family income level. A third major limitation is that the results were representative only for low- and middle-income Chilean preschool children and adolescents from southeastern Santiago and may not be generalizable to preschool children and adolescents from different income levels or from diverse Chilean regions. Nevertheless, both cohorts were similar with regard to the sociodemographic characteristics and nutritional status of the rest of the country [41]. In addition, we classified foods and beverages consumed at another person’s home with foods consumed at food courts, restaurants, cinemas, and other locations due to the small sample size of the latter set of eating locations. Because eating behaviors at others’ homes may differ from eating behaviors at other away-from-home locations, such as food courts or restaurants, future research will be needed to disentangle these locations. An additional limitation is that we classified some foods as mixed dishes based on standardized recipes, which may underestimate mixed dishes and homemade fast foods that were not part of the standardized recipes. Also, our food classification system was not able to distinguish between ready-to-eat dishes (e.g., frozen meals) and homemade dishes. Nevertheless, ready-to-eat dishes are not relevant calorie contributors in the Chilean diet, because they represent 1% of the total daily calories consumed [10]. Finally, we had only a single 24 h dietary recall, which may not be representative of usual intake. Nonetheless, a single 24 h dietary recall is a good method to estimate the mean intake at the group level [42].

Despite these limitations, this study differentiates home and away-from-home food sources and eating locations, and it is the first study to do so for Chilean preschool children and adolescents. Additionally, the dietary data were collected prior to the Chilean regulations and set an important baseline from which to evaluate the regulations.

## 5. Conclusions

Chilean preschool children and adolescents from Southeast Santiago obtained and ate most of their calories, saturated fats, total sugars, and sodium at home. At-home foods and beverages had lower caloric but higher sugar and sodium densities than foods obtained at away-from-home sources. Foods consumed at school had higher caloric, saturated fat, and total sugar densities than foods consumed at home. Most of the calories from SSBs were obtained at home, but the consumption of these beverages was relatively high across all eating locations. Future research is needed to understand whether Chile’s 2016 regulations that require warning labels; restrict marketing of foods and beverages high in calories, added sugars, saturated fats, and sodium; and ban regulated products from schools have affected dietary quality across food sources and eating locations.

## Figures and Tables

**Figure 1 nutrients-11-01695-f001:**
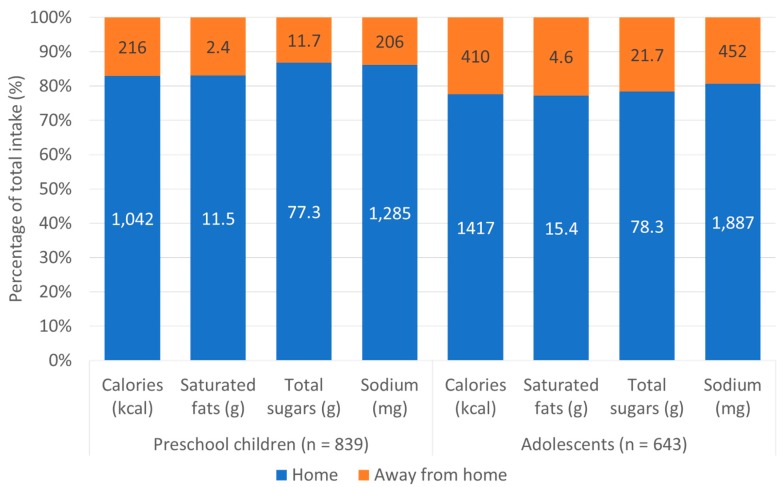
Percentage and absolute per capita intake of calories and key nutrients of concern by food source among low- and middle-income Chilean preschool children and adolescents from Southeast Santiago, 2016. Note: Values inside the bars represent the absolute mean intake by food source. We calculated percentages by dividing the total daily intake of each nutrient by the intake from each food source.

**Figure 2 nutrients-11-01695-f002:**
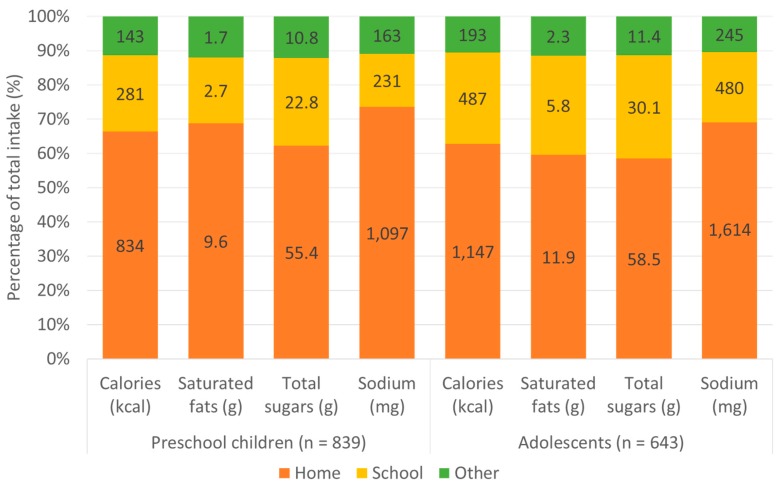
Percentage and absolute per capita intake of calories and key nutrients of concern by eating location among low- and middle-income Chilean preschool children and adolescents from Southeast Santiago, 2016. Note: Values inside the bars represent the absolute intake by eating location. We calculated percentages by dividing the total intake of each nutrient by the intake at each eating location.

**Table 1 nutrients-11-01695-t001:** Characteristics of the participants.

	Preschool Children (n = 839)	Adolescents (n = 643)
Age, years (mean, SE) *	4.3 (0.5)	13.2 (0.5)
Gender, female (%)	50.8	49.0
Mother’s education level (%)		
Less than high school	18.0	28.6
High school	40.6	45.6
More than high school	41.4	24.1
Missing	0.0	1.7
Calories, kcal/day (mean, SE)	1257 (13.6)	1826 (24.9)
Saturated fats, g/day (mean, SE)	13.9 (0.2)	20.0 (0.4)
% of calories from saturated fats (mean, SE)	9.9 (0.1)	9.7 (0.1)
Total sugars, g/day (mean, SE)	89.0 (1.3)	99.9 (2.1)
% of calories from total sugars (mean, SE)	28.6 (0.3)	21.9 (0.3)
Sodium, mg/day (mean, SE)	1491 (23)	2339 (41)
Sodium, mg/100 kcal (mean, SE)	130.6 (1.8)	119.7 (1.5)
Consumers reporting sources (%)		
Home	100.0	99.7
Away from home	59.2	72.3
Consumers reporting locations (%)		
Home	99.1	98.3
School	73.4	77.1
Other	45.8	35.6

* SE, standard error.

**Table 2 nutrients-11-01695-t002:** Nutrient density per 100 g and per 100 calories by food source * among low- and middle-income Chilean preschool children and adolescents from Southeast Santiago, 2016.

Preschool Children						Adolescents					
Nutrient	Home		Away from Home		*p*-value ^†^	Nutrient	Home		Away from Home		*p*-value ^†^
	Median	IQR ^‡^	Median	IQR			Median	IQR	Median	IQR	
Per 100 g						Per 100 g					
Calories, kcal	90.8	31.6	100.2	89.7	<0.001	Calories, kcal	105.7	43.9	131.5	120.4	<0.001
Saturated fats, g	0.9	0.6	0.9	1.2	0.718	Saturated fats, g	1.1	0.8	1.2	1.6	<0.001
Total sugars, g	6.7	3.5	4.3	6.5	<0.001	Total sugars, g	5.6	3.8	5.3	7.4	0.016
Sodium, mg	104.3	61.5	80.5	97.1	<0.001	Sodium, mg	133.4	79.6	115.4	135.3	0.904
Per 100 calories **						Per 100 calories **					
Saturated fats, %	9.6	4.5	8.9	6.4	0.738	Saturated fats, %	9.2	4.6	9.1	6.5	0.612
Total sugars, %	29.7	13.3	20.8	23.6	<0.001	Total sugars, %	21.9	13.7	19.2	23.5	0.308
Sodium, mg/100 kcal	116.2	55.4	85.0	49.8	<0.001	Sodium, mg/100 kcal	127.0	56.0	93.2	66.4	<0.001

* We determined the food source by combining foods purchased at groceries, convenience stores, or supermarkets and products made at home into one category (home) and combining ready-to-eat food obtained at school and other sources, such as restaurants and fast food establishments, into another category (away from home). ** For saturated fats and total sugars, we calculated the percentage of calories that each of these nutrients contributed to the total daily calories of each food source. For sodium we estimated the intake of sodium (mg) per 100 calories. ^†^ We assessed differences using the Wilcoxon signed rank test comparing home versus school in preschool children (n = 428) and adolescents (n = 445). ^‡^ IQR, interquartile range.

**Table 3 nutrients-11-01695-t003:** Nutrient density per 100 g and per 100 calories by eating location * among low- and middle-income Chilean preschool children and adolescents from Southeast Santiago, 2016.

Preschool Children									Adolescents								
Nutrient	Home		School		*p*-value ^†^	Other		*p*-value ^‡^	Nutrient	Home		School		*p*-value ^†^	Other		*p*-value ^‡^
	Med ^§^	IQR ^|^	Med	IQR		Med	IQR			Med	IQR	Med	IQR		Med	IQR	
Per 100 g									Per 100 g								
Calories, kcal	89.5	36.1	99.0	40.0	<0.001	87.0	126.6	0.918	Calories, kcal	103.0	46.7	126.9	78.7	<0.001	109.4	151.4	0.049
Saturated fats, g	1.0	0.7	0.9	0.9	0.017	0.7	1.7	0.043	Saturated fats, g	1.0	0.8	1.3	1.5	<0.001	1.1	2.5	0.135
Total sugars, g	5.8	3.4	9.3	6.8	<0.001	5.7	11.3	0.399	Total sugars, g	4.9	3.7	7.5	7.5	<0.001	5.9	8.2	0.008
Sodium, mg	110.2	70.3	74.9	58.2	<0.001	68.1	131.7	<0.001	Sodium, mg	137.9	85.1	112.5	106.2	<0.001	104.5	204.8	0.011
Per 100 calories **									Per 100 calories **								
Saturated fats, %	9.8	5.2	7.8	6.7	<0.001	9.0	9.5	0.092	Saturated fats, %	8.7	5.0	9.9	7.0	<0.001	9.7	7.1	0.860
Total sugars, %	26.3	14.0	37.8	30.6	<0.001	33.4	25.0	<0.001	Total sugars, %	20.1	15.1	24.8	20.6	<0.001	25.7	27.7	<0.001
Sodium, mg/100 kcal	121.7	64.4	74.4	49.6	<0.001	82.7	88.9	<0.001	Sodium, mg/100 kcal	135.0	57.9	86.5	60.2	<0.001	99.7	95.1	<0.001

* We designated the eating location as home if food was consumed at home, school if food was consumed at school, and other if food was consumed at another person’s home, at a food court, at a cinema, at a restaurant, in the street, on transportation, or at another location. ** For saturated fats and total sugars, we calculated the percentage of calories that each of these nutrients contributed to the total daily calories consumed at each eating location. For sodium we estimated the intake of sodium (mg) per 100 calories. ^†^ We assessed the differences using the Wilcoxon signed rank test comparing home with school in children (n = 612) and adolescents (n = 479). ^‡^ We assessed the differences using the Wilcoxon signed rank test comparing home with other in children (n = 287) and adolescents (n = 185). ^§^ Med, Median. ^|^ IQR, interquartile ranges.

**Table 4 nutrients-11-01695-t004:** Percentage of per capita total daily calories from food groups by food source (home, away from home) * among low- and middle-income Chilean preschool children and adolescents from Southeast Santiago, 2016.

Food Groups	Preschool Children (n = 839)						Adolescents (n = 643)					
	Mean Calories from Home			Mean Calories from Away-From-Home Sources			Mean Calories from Home			Mean Calories from Away-From-Home Sources		
	Absolute			Absolute			Absolute			Absolute		
	*Mean*	*SE ^†^*	*%*	*Mean*	*SE*	*%*	*Mean*	*SE*	*%*	*Mean*	*SE*	*%*
Cereal-based foods	146.8	5.2	14.1%	5.2	0.9	2.4%	352.0	10.5	24.9%	28.1	3.5	6.9%
Grain-based desserts	96.1	5.4	9.2%	18.8	2.5	8.7%	174.0	11.5	12.3%	54.6	5.3	13.3%
Sweets and non-grain-based desserts	31.7	3.1	3.0%	13.5	1.6	6.3%	38.2	4.1	2.7%	26.0	3.3	6.3%
Salty snacks	18.8	2.8	1.8%	8.1	1.5	3.8%	49.1	7.1	3.5%	23.4	3.7	5.7%
Meat, poultry, and meat substitutes	41.4	2.7	4.0%	2.3	0.9	1.0%	99.6	6.0	7.0%	13.0	2.1	3.2%
Dairy products and dairy substitutes	58.2	2.8	5.6%	6.2	1.1	2.9%	58.7	4.2	4.1%	10.2	1.6	2.5%
Fruits, vegetables, and mushrooms	51.7	2.4	5.0%	6.0	0.7	2.8%	49.6	3.1	3.5%	12.7	1.5	3.1%
Oils and fats	31.5	1.4	3.0%	0.2	0.1	0.1%	39.7	2.4	2.8%	2.2	0.7	0.5%
Coffee and tea	4.5	0.6	0.4%	0.1	0.1	0.1%	38.5	2.8	2.7%	1.4	0.4	0.3%
Fast foods	12.4	1.9	1.2%	10.8	2.0	5.0%	27.4	4.3	1.9%	21.8	4.3	5.3%
Sugar-sweetened beverages	169.5	5.4	16.3%	9.2	1.2	4.3%	153.4	6.5	10.8%	29.1	3.0	7.1%
Unsweetened dairy drinks and substitutes	57.7	4.4	5.5%	20.5	2.0	9.5%	24.6	2.7	1.7%	14.3	1.5	3.5%
Mixed dishes	143.7	6.1	13.8%	77.7	5.4	36.0%	166.1	9.8	11.7%	95.6	6.7	23.3%
Empanadas and sandwiches	8.6	2.6	0.8%	22.7	2.3	10.5%	36.5	6.8	2.6%	58.8	5.7	14.4%
Other food groups	169.0	6.1	16.2%	14.4	1.8	6.7%	109.0	5.3	7.7%	18.4	2.3	4.5%
Daily per capita total	1041.0	14.2	100.0%	216.0	10.1	100.0%	1417.0	25.5	100.0%	410.0	17.4	100.0%

* We determined the food source by combining food purchased at groceries, convenience stores, or supermarkets and products made at home into 1 category (home) and combining ready-to-eat food obtained at school, restaurants, fast food establishments, or other sources into another category (away from home). **^†^** SE, Standard error.

**Table 5 nutrients-11-01695-t005:** Percentage of per capita total daily calories from food groups by eating location (home, school, other) * among low- and middle-income Chilean children and adolescents from Southeast Santiago, 2016.

Food Groups	Preschool Children (n = 839)									Adolescents (n = 643)								
	Mean Calories Eaten at Home			Mean Calories Eaten at School			Mean Calories Eaten at other Locations			Mean Calories Eaten at Home			Mean Calories Eaten at School			Mean Calories Eaten at other Locations		
	Absolute			Absolute			Absolute			Absolute			Absolute			Absolute		
	*Mean*	*SE ^†^*	*%*	*Mean*	*SE*	*%*	*Mean*	*SE*	*%*	*Mean*	*SE*	*%*	*Mean*	*SE*	*%*	*Mean*	*SE*	*%*
Cereal-based foods	124.8	4.8	15.0%	13.2	1.4	4.7%	14.0	2.1	9.8%	311.0	10.0	27.1%	44.7	4.1	9.2%	24.4	3.5	12.7%
Grain-based desserts	56.1	4.4	6.7%	36.2	2.8	12.9%	22.6	2.8	15.8%	95.0	9.2	8.3%	107.0	6.9	22.0%	26.6	4.4	13.8%
Sweets and non-grain-based desserts	26.1	3.0	3.1%	6.5	1.1	2.3%	12.5	1.6	8.8%	29.7	3.8	2.6%	22.0	2.6	4.5%	12.4	2.4	6.5%
Salty snacks	11.8	2.4	1.4%	8.7	1.6	3.1%	6.4	1.2	4.5%	35.5	6.1	3.1%	21.9	3.7	4.5%	15.1	3.7	7.9%
Meat, poultry, and meat substitutes	35.8	2.6	4.3%	1.9	0.4	0.7%	6.0	1.3	4.2%	81.8	5.5	7.1%	17.4	2.3	3.6%	13.4	2.4	7.0%
Dairy products and dairy substitutes	38.1	2.3	4.6%	20.9	1.7	7.4%	5.4	1.0	3.8%	44.2	3.9	3.9%	18.6	2.0	3.8%	6.1	1.2	3.2%
Fruits, vegetables, and mushrooms	39.0	2.2	4.7%	13.3	1.0	4.7%	5.3	0.8	3.7%	40.4	2.9	3.5%	15.6	1.5	3.2%	6.3	1.2	3.3%
Oils and fats	27.6	1.3	3.3%	0.9	0.2	0.3%	3.2	0.6	2.2%	35.8	2.3	3.1%	3.8	0.7	0.8%	2.2	0.6	1.2%
Coffee and tea	4.2	0.6	0.5%	0.0	0.0	0.0%	0.4	0.2	0.3%	36.3	2.8	3.2%	1.3	0.4	0.3%	2.2	0.5	1.2%
Fast foods	13.1	2.0	1.6%	1.3	0.7	0.5%	8.7	1.8	6.1%	24.4	3.9	2.1%	6.6	1.9	1.4%	18.2	4.2	9.5%
Sugar-sweetened beverages	111.9	4.7	13.4%	46.2	2.3	16.5%	20.7	2.1	14.5%	113.5	5.7	9.9%	46.7	3.3	9.6%	22.3	2.8	11.6%
Unsweetened dairy drinks and substitutes	55.9	4.4	6.7%	20.1	1.4	7.2%	2.3	0.6	1.6%	23.3	2.6	2.0%	15.1	1.5	3.1%	0.5	0.4	0.3%
Mixed dishes	133.3	6.1	16.0%	71.3	5.1	25.4%	16.7	2.3	11.7%	143.5	9.1	12.5%	109.6	7.2	22.5%	8.5	2.1	4.4%
Empanadas and sandwiches	8.2	1.9	1.0%	17.8	1.9	6.3%	5.4	2.1	3.8%	37.6	6.6	3.3%	36.4	3.9	7.5%	21.3	4.7	11.1%
Other food groups	148.0	5.6	17.7%	22.3	1.9	8.0%	13.1	1.8	9.2%	94.7	5.1	8.3%	19.9	2.1	4.1%	12.8	2.2	6.6%
Daily per capita total	834.0	13.9	100.0%	281.0	9.9	100.0%	143.0	9.8	100.0%	1147.0	24.5	100.0%	487.0	16.7	100.0%	193.0	16.5	100.0%

* We designated the eating location as home if food was consumed at home, school if food was consumed at school, and other if food was consumed at another person’s home, at a food court, at a cinema, at a restaurant, in the street, on transportation, or at another location. **^†^** SE, Standard error.

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
