# Peer review of "Dietary Intake by Food Source and Eating Location in Low- and Middle-Income Chilean Preschool Children and Adolescents from Southeast Santiago"

_nutrients, 2019, doi:10.3390/nu11071695_

Round 1
Reviewer 1 Report
The authors have been highly responsive to the prior review; the paper has been improved. The authors did not elect to remove the word "critical" per Reviewer 2's recommendation but this should be fine.
Author Response
Dear reviewers and editorial board,
Thank you for the thoughtful comments on our paper. We have responded point by point and have made the corresponding changes to the manuscript. We feel that this represents a stronger version of these results and look forward to your further review. All the changes have been highlighted in yellow.
Comments from reviewers to authors:
Reviewer 2:
Please make sure that when using a term 'critical nutrients' you explain to what you refer to; for example iron can be called critical nutrient when we talk about proper brain development during pregnancy and infancy; similar folic acid will be a critical nutrient for a proper development of nervous system
Our response: Thank you for this important recommendation. We have replaced the term “critical nutrients” by “key nutrients of concern” in the whole manuscript and supplemental tables.
Line 100 - please clarify what you mean by "...linking foods and beverages by specific critical nutrients that food companies reported for packaged foods." Also what are specific critical nutrients?
Our response: Thank you for this relevant question. We have replaced these sentences with a simpler, clearer explanation of the linking system, which is simply that foods and beverages were linked at the food level to the most similar corresponding item in the USDA Food Composition Database (lines 99 - 100).
Lines 262 - 265 - where is this data shown? you do not refer to any table. "top calorie contributor" do you mean to the total daily energy? or to the total energy consumed at particular location?
Our response: Thank you for this important question. You are correct that these are data that do not appear in any of our tables. We have removed this paragraph from the results because the focus of the results and subsequent discussion is the intake of food groups by food source or eating location, not overall intake of food groups.

Reviewer 2 Report
Please make sure that when using a term 'critical nutrients' you explain to what you refer to; for example iron can be called critical nutrient when we talk about proper brain development during pregnancy and infancy; similar folic acid will be a critical nutrient for a proper development of nervous system
Line 100 - please clarify what you mean by "...linking foods and beverages by specific critical nutrients that food companies reported for packaged foods." Also what are specific critical nutrients?
Lines 262 - 265 - where is this data shown? you do not refer to any table. "top calorie contributor" do you mean to the total daily energy? or to the total energy consumed at particular location?
Author Response
Dear reviewers and editorial board,
Thank you for the thoughtful comments on our paper. We have responded point by point and have made the corresponding changes to the manuscript. We feel that this represents a stronger version of these results and look forward to your further review. All the changes have been highlighted in yellow.
Comments from reviewers to authors:
Reviewer 2:
Please make sure that when using a term 'critical nutrients' you explain to what you refer to; for example iron can be called critical nutrient when we talk about proper brain development during pregnancy and infancy; similar folic acid will be a critical nutrient for a proper development of nervous system
Our response: Thank
you for this important recommendation. We have replaced the term
“critical nutrients” by “key nutrients of concern” in the whole
manuscript and supplemental tables.
Line 100 - please clarify what you mean by "...linking foods and beverages by specific critical nutrients that food companies reported for packaged foods." Also what are specific critical nutrients?
Our response: Thank you for this relevant question. We have replaced these sentences with a simpler, clearer explanation of the linking system, which is simply that foods and beverages were linked at the food level to the most similar corresponding item in the USDA Food Composition Database (lines 99 - 100).
Lines 262 - 265 - where is this data shown? you do not refer to any table. "top calorie contributor" do you mean to the total daily energy? or to the total energy consumed at particular location?
Our response: Thank you for this important question. You are correct that these are data that do not appear in any of our tables. We have removed this paragraph from the results because the focus of the results and subsequent discussion is the intake of food groups by food source or eating location, not overall intake of food groups.

This manuscript is a resubmission of an earlier submission. The following is a list of the peer review reports and author responses from that submission.
Round 1
Reviewer 1 Report
The article describes the children and adolescents daily intake of critical nutrient. I don't have any comment
Author Response
We appreciate your feedback.
Reviewer 2 Report
Please see attached comments.
nutrients-525183
Where are low- and middle-income Chilean preschool children and adolescents getting their critical nutrients? Dietary intake by food source and eating location
Thank you for the opportunity to review this manuscript. This is an interesting study, especially considering 2016 regulations related to food marketing in Chile. Results from this study may be applicable to other researchers or health professionals interested in diet quality and location of intake in Chilean children. The manuscript is very well written and provides a nice summary of the study and results. Although the research is important, the manuscript is missing some methodological details that could enhance the paper. Please see specific comments below.
1. The sample was drawn from southeastern Santiago but the title and much of the manuscript imply that this was a national sample. This is noted as a limitation in the Discussion section but needs to be clear throughout the paper.
2. The authors note that the sample is composed of low- and middle-income children. However, does this information stem from neighborhood of residence only or based on report of actual family income? If the former, this should be clear and noted as a limitation.
3. For the FECHIC cohort, mothers were excluded if they had “any mental disability”. How was this defined and ascertained?
4. Did children ages 12 to 14 years who reported their own dietary intake also assent to participate in the study?
5. Page 3, lines 94-96 – how often was a Chilean food or beverage not in the USDA Food Composition Databases and had to be linked to a US food or beverage nutrient profile.
6. Please provide information on how many children had a second 24-hour recall by study rather than overall.
7. At what age do children start school in Chile. Assuming this is age 4 or 5, were some children in nursery and some in school at the time of assessment? Were these settings both classified as “school”? Were these settings examined separately in sensitivity analyses? This information should be provided in the manuscript.
8. Please provide more information on why foods and beverages consumed at another person’s home (e.g., grandparent) were classified with food courts, cinemas, restaurant, etc.? One could assume that these foods and beverages consumed at another’s home might look like those consumed in the child’s home (vs. at a restaurant, for example). Did the authors conduct sensitivity analyses?
9. Please provide additional justification for removing outliers of saturated fat and sodium intake at the 99.5 percentile.
10. Minor comment: please be mindful of changing tenses when presenting study results in the Discussion section.
11. Minor comment: having only a single recall should be noted as a limitation in the Discussion section.
12. Minor comment: the title could be modified – are both sentences really needed when they are somewhat redundant?
Author Response
Dear reviewers and editorial board,
Thank you for the thoughtful comments on our paper. We have responded point by point and have made the corresponding changes to the manuscript. We feel that this represents a stronger version of these results and look forward to your further review.
Please see attachment.

Reviewer 3 Report
Dear Authors,
It was nice to read your manuscript. Please find some comments/suggestions below:
Page 1 Line 21 - please remove the word 'critical' (critical is rather a vague term; remove this term throughout the whole manuscript) and specify which nutrients in brackets
Page 1 line 22 -23 - sentence starting with 'However, ...' indicates that it is positive that that food obtained or eaten at school or other away-from-home locations is higher in energy, SFAs, sugar; moreover this sentence needs to be rewritten as it is not clear if you are referring to location or source; also the sentence doe not seem to be supported with the results presented in the main text; for example there were differences between children and adolescents
Abstract - conclusion - needs to be rewritten to more emphasise the findings of this study
Line 307 - please look at the comment for the line 22
Line 311 - what are 'cereal-based desserts' have not been shown in the results section
Line 312 - looking at the results it seems that more sugar-sweetened beverages (SSB) are consumed at home rather than at school
Line 324 - "...came from similar groups" - not clear similar to what groups?
Line 352 - not really 'school food' but food consumed at school; do not mix the location with the source
Line 397-399 - this sentence needs to be rewritten as your point is not clear
Line 395 - comment as for the line 22
Line 400 - your results do not necessarily support this statement; you have shown the absolute energy and selected nutrients intake but you have not compared it to the recommended intake, so we do not really know if their diet is proper or inadequate
Table 2 and 3- I recommend showing Q1 and Q3
Table 3 - I recommend using the Kruskal-Walis test to compare all locations followed by post hoc analysis
Author Response

(The authors gave the same response as above.)
